# Validated LC-MS/MS Assay for the Quantitative Determination of Fenretinide in Plasma and Tumor and Its Application in a Pharmacokinetic Study in Mice of a Novel Oral Nanoformulation of Fenretinide

**DOI:** 10.3390/pharmaceutics16030387

**Published:** 2024-03-12

**Authors:** Cristina Matteo, Isabella Orienti, Adriana Eramo, Ann Zeuner, Mariella Ferrari, Alice Passoni, Renzo Bagnati, Marianna Ponzo, Ezia Bello, Massimo Zucchetti, Roberta Frapolli

**Affiliations:** 1Laboratory of Cancer Pharmacology, Department of Oncology, Istituto di Ricerche Farmacologiche Mario Negri IRCCS, 20156 Milan, Italy; cristina.matteo@marionegri.it (C.M.); mariella.ferrari89@gmail.com (M.F.); mariannaponzo@virgilio.it (M.P.); ezia.bello@marionegri.it (E.B.); roberta.frapolli@marionegri.it (R.F.); 2Department of Pharmacy and Biotechnology, University of Bologna, 40126 Bologna, Italy; isabella.orienti@unibo.it; 3Department of Oncology and Molecular Medicine, Istituto Superiore di Sanità, 00161 Rome, Italy; adriana.eramo@iss.it (A.E.); ann.zeuner@iss.it (A.Z.); 4Department of Environmental Health Sciences, Istituto di Ricerche Farmacologiche Mario Negri IRCCS, 20156 Milan, Italy; alice.passoni@marionegri.it (A.P.); renzo.bagnati@marionegri.it (R.B.)

**Keywords:** fenretinide, 4-HPR, oral formulation, pharmacokinetics, tumor distribution, analytical chemistry

## Abstract

We describe the development and validation of a HPLC-MS/MS method to assess the pharmacokinetics and tumor distribution of fenretinide, a synthetic retinoid chemically related to all-trans-retinoic acid, after administration of a novel oral nanoformulation of fenretinide, called bionanofenretinide (BNF). BNF was developed to overcome the major limitation of fenretinide: its poor aqueous solubility and bioavailability due to its hydrophobic nature. The method proved to be reproducible, precise and highly accurate for the measurement of the drug and the main metabolites. The lower limit of quantification resulted in 1 ng/mL. The curve range of 1–500 ng/mL and 50–2000 ng/mL, for plasma and tumor homogenate, respectively, was appropriate for the analysis, as demonstrated by the accuracy of between 96.8% and 102.4% for plasma and 96.6 to 102.3% for the tumor. The interdays precision and accuracy determined on quality controls at three different levels were in the ranges of 6.9 to 7.5% and 99.3 to 101.0%, and 0.96 to 1.91% and 102.3 to 105.8% for plasma and tumor, respectively. With the application of the novel assay in explorative pharmacokinetic studies, following acute and chronic oral administration of the nanoformulation, fenretinide was detected in plasma and tumor tissue at a concentration higher than the IC50 value necessary for in vitro inhibitory activity (i.e., 1–5 µM) in different cancer cells lines. We were also able to detect the presence in plasma and tumor of active and inactive metabolites of fenretinide.

## 1. Introduction

Fenretinide (Figure 1), *N*-(4-hydroxyphenyl)retinamide (4-HPR) is a synthetic retinoid chemically related to all-trans-retinoic acid (ATRA), the acidic form of vitamin A [1].

The main mechanism of action of 4-HPR is the induction of apoptosis rather than cellular differentiation, which in contrast, is mainly induced by ATRA. 4-HPR induces tumor cell apoptosis through the generation of radical oxygen species, the imbalance of ceramides/dehydroceramides ratio and the induction of retinoic acid receptor β. 4-HPR can also induce antiangiogenic and antimetastatic effects as demonstrated in several tumor models. These peculiar characteristics have made 4-HPR currently the most studied retinoid both as a chemopreventive and chemotherapeutic agent [2,3].

4-HPR has been evaluated in solid tumors and hematological malignancies in several clinical trials [4] which demonstrated its excellent tolerability but a limited therapeutic efficacy due to its poor bioavailability. In fact, the scant aqueous solubility of 4-HPR restrains its absorption thus preventing the achievement of plasma concentrations suitable to elicit a therapeutic response. The clinical studies with 4-HPR have been mainly conducted via conventional formulations such as soft gelatin capsules, available at the National Cancer Institute, containing 100 mg 4-HPR of corn oil and polysorbate 80. Multiple and protracted administrations of 4-HPR via the capsules provided low plasma concentrations of the drug, always below the minimum threshold required for therapeutic activity [4], demonstrating the unsuitability of this formulation to raise drug bioavailability to levels within the therapeutic window. To improve 4-HPR bioavailability, nanofenretinide, a new formulation of fenretinide complexed with 2-hydroxypropyl-beta-cyclodextrin, was delivered intravenously and showed effectiveness against multiple solid tumors including lung and colorectal cancers [5]. Then, an improved oral formulation was developed, named bionanofenretinide (BNF), based on drug complexation with a mixture of phospholipids and triglycerides providing nanocapsules. BNF is characterized by high drug loading, high aqueous solubility and increased oral absorption [6].

To study the new formulation during preclinical investigation, it was necessary to set up and validate a method to measure 4-HPR concentrations in plasma and tumors. In the present paper we reported the new methodology developed and applied during both pharmacokinetic and activity studies performed with BNF [6]. Despite several methods based on mass spectrometry or traditional HPLC exist for the determination of retinoids in plasma [7,8,9,10,11], only a few methods for the specific quantitation of 4-HPR are available [10,12,13,14]. To achieve a specific, sensitive and rapid procedure for the quantification of fenretinide in plasma and tumor, we developed and validated a high-performance tandem liquid chromatography-mass spectrometry (HPLC-MS/MS) method. It requires only 30 µL of plasma, a small amount of tumor homogenate, a simple solvent treatment and a short analysis time. High selectivity and sensitivity were ensured by operating in multiple reaction monitoring (MRM) mode. Differently from the existing methods, the presented method is able to quantify a wide range of 4-HPR concentrations (from 1 ng/mL to 2000 ng/mL). Moreover, 4-HPR metabolites were identified via high-resolution mass spectrometry (HRMS), using an Orbitrap instrument operating in ESI positive ion mode. Subsequently, the transitions identified throughout HRMS were used for the quantitation of metabolites.

The present method was successfully applied in a pharmacokinetic study in mice, also highlighting the tumor distribution of BNF and providing for the first time, data on its metabolism.

## 2. Materials and Methods

### 2.1. Reagents and Chemicals

Fenretinide (*N*-(4-hydroxyphenyl) retinamide), was purchased from Olon Spa (Milan, Italy). Deuterated fenretinide (N(-4-hydroxyphenyl-d4) retinamide), C_26_H_29_D_4_NO_2_ ([^2^H_4_]-4-HPR), used as internal standard (IS), was obtained from Tocris Bioscience (Bristol, UK).

HPLC-grade methanol and acetonitrile were purchased from Carlo Erba (Milan, Italy) and analytical grade formic acid (98%) from Sigma-Aldrich Co. (Milan, Italy). Deionized water was prepared using a Milli-Q water purifying system from Millipore Corp. (Bedford, MA, USA).

### 2.2. Animals

Female CD1 mice, 7 weeks old, were supplied by Envigo RMS SRL (Udine, Italy). Animals were housed in the institute’s animal care facilities, which meet international standards; they were regularly checked by a certified veterinarian who was responsible for health monitoring, animal welfare supervision, experimental protocols and procedures revision.

### 2.3. Preparation of Standard and Quality Control Solutions and Samples

Two separate ethanol stock solutions of 4-HPR for the preparation of standards and quality controls (QCs), necessary to the assay of plasma and tumor samples, were prepared at 1 mg/mL and further diluted in acetonitrile to obtain the appropriate working solutions of 10 to 20,000 ng/mL. The stock solution for IS was prepared in ethanol at 1 mg/mL and then diluted to working solutions of 300 ng/mL and 3200 ng/mL for plasma and tumor, respectively. All stock and working solutions were stored in the dark at −20 °C until use.

To assay plasma, eight calibration standards and three levels of QCs samples were prepared by adding 10 µL of different working solutions to 90 µL of control murine plasma to obtain final standard concentrations of 1, 5, 10, 25, 50, 100, 250 and 500 ng/mL and 8, 80, 400 ng/mL for QCs (QL, QM, QH). 

To validate the quantification method of 4-HPR in tumor, a six-point calibration curve was prepared for each analytical session by adding 15 µL of working solution of standards or QC to 135 µL of tumor control homogenate; specifically, an A2780 ovarian cancer model was used. Final 4-HPR standards at 50, 100, 250, 500 1000 and 2000 ng/mL, corresponding to tissue concentrations of 350, 700, 1750, 3500, 7000 and 14,000 ng/g and QCs of 150, 750 and 900 ng/mL, corresponding to tissue concentrations of 1050, 5250 and 6300 ng/g were obtained.

### 2.4. Preparation of Plasma Samples

After thawing plasma samples at room temperature, an aliquot of 30 µL was transferred to a 1.5 mL Eppendorf polypropylene tube, spiked with 3 ng (10 µL) of IS and diluted with 90 µL of acetonitrile to deproteinize plasma. Each tube was thoroughly vortexed for 30 s, shaken for 5 min at 1250 rpm and centrifuged for 5 min at 4000× *g*. The obtained supernatant was then transferred to an amber glass vial and 10 µL were injected into the HPLC-MS/MS system. Amber glass vials and aluminum foil were used as precautions to minimize exposure of the analytes to the light to avoid photodegradation.

### 2.5. Preparation of Tumor Homogenate Samples

Tumors were generated via subcutaneous injection of cancer stem cells from lung cancer (LCSC-136) and melanoma (MEL 3) cell lines. Since 4-HPR is highly lipophilic, acetonitrile was used for tumor homogenization. Removed or control tumor was weighed, mixed with acetonitrile in a ratio of 1:6 (*w*/*v*) and homogenized (1 min) using an Ultra-Turrax (IK A, Staufen, Germany). An aliquot of 150 µL of tumor homogenate was spiked with 10 µL of IS working solution, mixed and centrifuged at 4000× *g* for 10 min at 4 °C. The supernatant was transferred to an amber glass vial and 10 µL was injected into the HPLC-MS/MS system.

### 2.6. Liquid Chromatographic Conditions

The HPLC system consisted of a Series 200 autosampler and micropump (Perkin Elmer, Waltham, MA, USA) with an online vacuum degasser and a temperature-controlled (32 °C) column compartment. Chromatographic separation was achieved via a Gemini-C18 column (50 mm × 2.0 mm, 5 µm particle size; Phenomenex Inc., Torrance, CA, USA) protected by Security Guard™ ULTRA cartridges C18 (Phenomenex Inc., Torrance, CA, USA). The mobile phases consisted of 0.05% formic acid in water (MP-A) and 100% methanol (MP-B). The chromatographic separation was performed at a flow rate of 0.3 mL/min, applying the following gradient steps: from 32 to 2% MP-A for 3 min; 2% MP-A held for 2 min; from 2 to 32% MP-A for 30 s and, as the last step, re-equilibration in the initial condition for 4 min. The autosampler was maintained at 4 °C and the injected volume was 10 µL.

### 2.7. Mass Spectrometry Conditions

Mass spectrometric detection was carried out on a triple quadrupole API 4000 mass spectrometer (Sciex, Framingham, MA, USA) equipped with an atmospheric pressure chemical ionization source (APCI) operating in positive ion mode at 350 °C, with a needle current of 4 µA. The nebulizer gas (Gas 1), heater gas (Gas 2), curtain gas (CUR) and collision activated dissociation gas (CAD) were set to 40, 50, 30 and 5 instrument units, respectively. Declustering potential (DP) was set at 60 V and the collision exit potential (CXP) at 15 V. All source parameters were optimized via direct infusion of 4-HPR under LC conditions. The quantification of 4-HPR and IS was carried out in MRM mode, using the pseudo-molecular (Q1) to fragment (Q3) ion transitions and the optimal collision energy (CE) reported in the following scheme.



**Analyte**

**Q1**

**(*m*/*z*)**

**Q3**

**(*m*/*z*)**

**CE**

**(eV)**
4-HPR392.3283.3164-oxo-4-HPR406.2297.1164-MPR406.3283.216DH-4-HPR390.2281.116[^2^H_4_]-4-HPR396.3283.218


The scheme above also lists the specific transition used to quantify 4-HPR metabolites, which were previously identified using a high-resolution LTQ-Orbitrap XL mass spectrometer (Thermo Scientific Inc., Waltham, MA, USA), equipped with an electrospray source (ESI) operating in positive ion mode (Figure 1). Chromatographic separation was performed using a 1200 series pump and auto sampler (Agilent Technologies, Santa Clara, CA, USA), with a Gemini-C18 column and a mobile phase composed of 0.1% formic acid in water (MP-A) and acetonitrile (MP-B). The injection volume was 8 µL and the flow rate was 200 µL/min. The elution gradient was from 2% to 99% of MP-B for 28 min, held to 99% of MP-B for 2 min and then re-equilibration at 2% of MP-B for ten minutes was performed. Full scan MS spectra were acquired in the *m*/*z* range 100–800 (60,000 resolving power), while MS/MS spectra were acquired in the *m*/*z* range 50–500 with a collision energy of 28 eV (15,000 resolving power).

### 2.8. Validation Procedures

Method validation was performed according to the European Medicines Agency and the Food Drug and Administration guidelines on bioanalytical method validation [15,16]. These methods were validated in terms of linearity, carry-over, intra and interday precision and accuracy, lowest limit of quantification (LLOQ), selectivity, matrix effect and recovery. Moreover, stability tests were performed both in plasma and in tumor homogenate.

#### 2.8.1. Limit of Quantification, Matrix Effect and Recovery

Six different batches of control plasma and different tumor types were spiked with 4-HPR at the LLOQ level of 1 ng/mL and 50 ng/mL (corresponding to 350 ng/g in tissue), respectively, to investigate the selectivity of the method. As defined by the guidelines of the main regulatory agencies, the LLOQ precision must be ≤20% and accuracy in the range 80–120% of the nominal value. During the preliminary phase of method development, we noticed that the actual concentrations found in samples from treated mice were quite superior to the instrumental detection limit (i.e., 0.5 ng/mL), so we decided to validate a LLOQ of 1 ng/mL for plasma and 50 ng/mL for tumor homogenate.

On the same independent sources of plasma and tumors, matrix effects and recovery were also investigated, analyzing the lowest 4-HPR QC concentrations.

Matrix effect was calculated by evaluating the normalized matrix factor. Matrix factor (MF) is the ratio between the peak area of analyte in spiked matrix and the peak area in the absence of matrix (pure solution of the analyte). A value of MF close to 1 defines absence of matrix effect. 

The normalized matrix factor is the ratio of the analyte MF to the internal standard MF, a value close to 1 defines the absence of matrix effect. The normalized MF coefficient of variation (CV), calculated as a percentage ratio of the standard deviation to the mean calculated concentration, had to be lower than 15%.

The extraction efficiency of 4-HPR (recovery) was determined by comparing the peak area of analyte extracted from plasma or tissue homogenate QC samples with the peak area of the extracted matrix samples containing the same amount of analyte added following the extraction procedures. IS recovery was determined in the same way at the concentration of 30 ng/mL; 200 ng/mL CV% had to be within 15%.

#### 2.8.2. Linearity

The linearity of the standard calibration curve between 1 to 500 ng/mL for plasma and between 50 to 2000 ng/mL in tumor tissue was evaluated during different analytical runs using fresh preparations. Each calibration curve consisted of a double blank sample, a blank sample and eight calibrators for plasma or six for tumor homogenate. The double blank sample was reinjected also after the highest concentration standard in each run to monitor the 4-HPR and IS carry-over. 

The ratio between the peak area HPLC-MS/MS for 4-HPR and the IS (y) was determined for each standard point and plotted against the nominal concentration of 4-HPR in the sample (x) using a weighted (1/x^2^) least squares linear regression analysis. The acceptance criteria for accuracy of the back-calculated values of each standard point had to be in the range of 85–115% of the nominal concentration and the LLOQ in the range 80–120%. At least 75% of the standard points of the calibration curve must meet the described criteria.

#### 2.8.3. Accuracy and Precision

Intraday precision and accuracy over one day was checked by measuring the analytes in five replicates of LLOQ and QC samples. The interday precision and accuracy on different days were checked by measuring the analytes in three replicates of QC samples. The LLOQ and QC levels were analyzed at nominal concentrations of 1, 8, 80 and 400 ng/mL for plasma and 50, 150, 750 and 900 ng/mL for tumor. To analyze the QCs, we prepared and processed a fresh standard calibration curve for each analytical run. The precision of the method was determined using CV% and the accuracy expressed as the percentage ratio between the calculated mean concentration and the nominal concentration. 

For each QC, the measured concentration must be within 15% of the nominal value in at least 67% of QCs samples (2/3). 

The ability to dilute plasma sample originally above the upper limit of quantification (ULOQ) was assayed by analyzing QC samples containing 4 and 10 times the concentration of the high QC sample, after a 4-fold and a 10-fold dilution in control murine plasma.

#### 2.8.4. Stability

The stability of 4-HPR in plasma and tumor homogenate was assessed by analyzing QC samples in triplicate during storage and handling. In particular, the bench-top stability was determined after 4 h at room temperature, the long-term stability after 4 months of storage for plasma and 3 weeks of storage for tumor homogenate at −20 °C and the stability in the auto-sampler by reanalyzing the processed QC samples 24 h after the first injection. The drug was considered stable in each QC concentration when differences between the freshly prepared and stability tested samples did not differ more than 15 percent from the nominal concentrations.

#### 2.8.5. Application to the Pharmacokinetic Study

In this paper, we reported the results obtained in the first exploratory pharmacokinetics carried out to test the performance of the method, which allowed us to obtain preliminary information on the bioavailability of BNF. The oral and intravenous pharmacokinetics were investigated in CD1 female mice (mean weight 25 ± 1 g), 7 weeks of age, divided into two groups of 24 mice, randomized to receive a single dose of BNF at the 4-HPR equivalent dose of 5 and 10 mg/kg in comparison with free 4-HPR. BNF was dissolved in sterile water while free 4-HPR, due to its water insolubility, was dissolved in the same mixture of corn oil and polysorbate 80 (0.92:0.08, *w*:*w*) as contained in the soft gelatin capsules, for oral administration or in ethanol for IV administration. Both BNF and free 4-HPR were administered via gavage or bolus injection. After treatment, a series of blood samples were taken at 0.08 (iv) or 0.25 (po), 0.5, 1, 2, 4, 10, 24 and 48 h (3 mice for time point). Blood was collected in heparinized tubes from the retro-orbital plexus of the mice under isoflurane anesthesia. To obtain plasma (about 250–300 µL), blood samples were centrifuged at 4000 rpm for 10 min at 4 °C. All the collected samples were stored at −20 °C until analysis.

The obtained 4-HPR concentration versus-time data were elaborated via noncompartmental analysis by means of PKSolver, an add-in program for pharmacokinetic data analysis in Microsoft Excel (LTSC Professional Plus 2021) [17].

In a different experiment of antitumor activity in mice growing melanoma (MEL 3) and lung cancer (LCSC-136) xenografts, the intratumor 4-HPR concentration was determined after repeated oral treatments with BNF or free 4-HPR at the 4-HPR equivalent dose of 100 mg/kg twice daily. 

## 3. Results

### 3.1. Method Development and Validation

#### 3.1.1. HPLC-MS/MS

Following the direct infusion of standard solution using both APCI and ESI source in positive ion mode, we decided to use the APCI source since we obtained a better and more stable signal of the analyte and IS. 4-HPR and [^2^H_4_]-4-HPR mainly formed a protonated pseudo-molecular molecule [M+H]^+^ at *m*/*z* 392.3 and 396.3, respectively. These precursor ions flowed through the first quadrupole in the collision cell and the CE was optimized to provide the product ions with the highest signal. Characteristic product ions were monitored in the third quadrupole at *m*/*z* 283.2 (16 eV) and 161.1 (25 eV) for both 4-HPR and [^2^H_4_]-4-HPR.

Representative MRM chromatograms of extracted plasma and tumor samples are shown in Figure 2 and Figure 3. The panels A, B and C refer respectively to double blank, blank and LLOQ samples at 1 ng/mL for plasma and 50 ng/mL (i.e., 350 ng/g) for tumor homogenate. Panel D shows the extracted plasma and tumor samples of a mouse treated daily for three weeks with 100 mg/kg of BNF and sacrificed 2 h after the last dose. The measured concentrations of 4-HPR were 1669 ng/mL in plasma and 1774 ng/g in tumor.

#### 3.1.2. LLOQ, Matrix Effect and Recovery

As explained in Section 2, we decided to validate the concentration of 1 ng/mL as LLOQ in plasma and 50 ng/mL in tumor homogenate. At this concentration, the precision and the accuracy were 7.6% and 107.6%, respectively, for plasma and 5.1% and 104.8% for tumor. 

The matrix effect was calculated at the low QC concentration level. The normalized matrix factor ranging from 0.96 to 1.15 for the six sources of plasma samples and from 0.97 to 1.03 for tumor homogenate. The calculated normalized matrix factors, close to one, indicated that no coeluting substances of the matrices affected the analyte signal. In particular, considering the tumor matrix, the procedure of homogenization in a large volume of solvent, together with an appropriate chromatographic separation, allowed for the achievement of satisfactory results.

We used acetonitrile for both plasma deproteinization and tumor homogenization. The recoveries, evaluated by processing five replicates at low and high QC concentrations, were: 85.5% (CV 3.0%) for QL and 98.7% (CV 11.8%) for QH in plasma and 93.8% (CV 5.8%) for QL and 94.9% (CV 7.6%) for QH in tumor. The IS recovery were 94.0% (CV 6.1%) and 103.2 (CV 8.6%) for tumor and plasma, respectively.

#### 3.1.3. Linearity

Table 1 reports the 4-HPR standard calibration curves in plasma and tumor, prepared on different days during the validation study. These calibration curves were generated by plotting the peak area ratios of the analyte to the IS versus the 4-HPR nominal concentrations and applying weighted quadratic regression function (1/x^2^). The plasma and tumor calibration curves were linear over the concentration ranges of 1–500 ng/mL and 50–2000 ng/mL, respectively. The back-calculated concentration accuracy was ≥96.8% in plasma and ≥96.6% in tumor with a precision, expressed as CV%, ≤4.5 and ≤3.4%. The Pearson’s coefficients of determination, R^2^, were ≥0.9966 for both plasma and tumor.

Data for carry-over evaluation were obtained in analytical runs injecting blank samples following the injection of the ULOQ. No signal of 4-HPR and of IS was observed during this analysis.

#### 3.1.4. Accuracy and Precision

Intraday accuracy and precision of the method were evaluated by analyzing five replicates of LLOQ and QC samples at 1, 8, 80 and 400 ng/mL for plasma and 50, 150, 750, 900 ng/mL for tumor homogenate, within a single-run analysis. The method appeared accurate and extremely precise, in fact, as shown in Table 2 and Table 3, the accuracy and precision comprised the ranges of 89.7 to 104.6% and 2.1 to 5.5% in plasma and 103.3 to 107.0% and 1.0 to 5.0% in tumor.

Satisfactory results were also obtained when the interday variability was determined in QC samples. As shown in Table 2 and Table 3, the precision and accuracy assessed in triplicate samples, over at least four days of analysis showed a range of 6.9 to 7.5% and 99.3 to 101.0% in plasma and 1.9 to 3.2% and 102.3 to 105.8% in tumor.

No dependence from dilution was observed in the analysis of 4-HPR in plasma, with the mean accuracy of the found concentrations of 86.7% and 88.0%, respectively; the precision was 1.41% and 1.78% for the dilution ratio 1:4 and 1:10, respectively.

#### 3.1.5. Stability

In order to evaluate 4-HPR’s stability in plasma and in tumor, QC samples were analyzed: three replicates at 8, 80 and 400 ng/mL for plasma samples and three replicates at 150, 750, 900 ng/mL for tumor samples. 

As a result, in plasma, 4-HPR remained stable after 4 months at −20 °C resulting in the ranges of 98.0 to 101.9% of the nominal concentrations. In tumor homogenate, 4-HPR remained stable at −20 °C for three weeks being 98.1 to 112.6% of the nominal concentrations. In both cases, the coefficient of variation was ≤10.0%. 

Bench-top stability (4 h at room temperature) and autosampler stability (24 h at 4 °C) for matrices was assessed to cover the preparation and injection period of analysis. We obtained an accuracy in the ranges of 106.1 to 108.4% (CV 1.2–3.5%) and 97.1 to 100.6% (CV 3.1–4.6%) for bench-top stability in plasma and tumor, respectively. In the autosampler, samples remained stable for 24 h with an accuracy of 92.1 to 97.6% (CV 4.0–6.7%) and 104.1 to 110.0% (CV 0.8–3.0%) for plasma and tumor, respectively.

### 3.2. Identification of 4-HPR Metabolites

Metabolites of 4-HPR were identified using a high-resolution mass spectrometer (LTQ-Orbitrap XL), which facilitated the identification of chemical structures. The analyses were carried out in plasma and tumor samples extracted as described above. The chemical structures of the identified metabolites and the corresponding MS/MS spectra used for the identification are shown in Appendix A. High resolutions MS/MS spectra allowed for the identification of three main metabolites and their principal fragments, which were then used for the MRM quantitative analysis. They corresponded to: *N-*(4-methoxyphenyl) retinamide (4-MPR), 4-oxo-*N*-(4-hydroxyphenyl) retinamide (4-oxo-4-HPR) and dehydrogenated 4-HPR (DH-4-HPR). 

### 3.3. Application to the Pharmacokinetic Study

The described method was successfully applied in an explorative pharmacokinetic study with BNF, in mice treated via oral gavage and intravenous bolus with 10 and 5 mg/kg of 4-HPR dose equivalent, respectively. Figure 4 shows the profiles of the measured plasma concentration-versus-time of 4-HPR obtained after BNF and for comparison the profile after free 4-HPR administered intravenously at 5 mg/kg.

From a visual inspection of the curves, it can be seen that the two intravenous profiles were superimposable and via oral BNF, 4-HPR achieved plasma Cmax between 2 and 4 h. It was distributed rapidly and eliminated with a half-life (HL) of about 6–7 h, warranting measurable drug plasma levels up to 48 h. From the comparison of the calculated experimental AUC, the bioavailability of BNF resulted in a value of 25% (AUC_0–last_ 4-HPR iv: 18.23 h*µg/mL; AUC_0–last_ BNF iv: 13.56 h*µg/mL; AUC_0–last_ BNF po: 6.745 h*µg/mL).

We subsequently studied the pharmacokinetics of BNF compared with free 4-HPR in one of the orally repeated treatment schedules employed during activity studies. The treatment consisted of doses of 100 mg/kg, given twice a day, for one week. Figure 5 reports the obtained 4-HPR plasma concentration profile also showing the measured metabolites of 4-HPR. 

The relative percent amounts of the active metabolite 4-oxo-4-HPR and the inactive metabolite MPR, versus 4-HPR, corresponded to about 50% and 17%, respectively. A third metabolite, DH-4-HPR, previously described by Cooper et al. [4] amounted to 5% 4-HPR (Table 4).

### 3.4. Application to the Activity Study

The method was also successfully applied to measurements of 4-HPR and its metabolites in xenografts generated from subcutaneous injection of MEL3 and LCSC-136 stem cells [6]. We measured mean 4-HPR concentrations of 2126 (5.4 µM) and 2228 ng/g (*N* = 3), 2 h after dosing. Interestingly, these concentrations were higher than the IC50 value necessary for in vitro inhibitory activity (i.e., 1–5 µM), therefore potentially effective. However, even more interestingly, thanks to the developed method, we discovered that 4-oxo-4-HPR, the active metabolite, was present in the tumor tissues in quantities comparable to 4-HPR, probably contributing to the overall antitumor activity triggered by BNF administration.

## 4. Discussion

Limited availability of methods for the determination of retinoids in plasma and tissues prevent us from fully understanding their effects, the correlations between concentration levels and activity in the different body compartments and the role of the pharmaceutical formulations in influencing their bioavailability and pharmacokinetics. 

In this study, we described the development of an accurate and reproducible bioanalytical method for the assessment of 4-HPR in plasma and tumor tissue. The method proved to be reproducible, precise and highly accurate with interday precision and accuracy determined by quality controls in the ranges of 6.9 to 7.5% and 99.3 to 101.0% and 1.9 to 3.2% and 102.3 to 105.8% for plasma and tumor, respectively.

The APCI source and MRM mode together ensured the high selectivity and sensitivity of the HPLC-MS/MS method. Differently from the existing methods, the presented method was able to quantify a wide range of 4-HPR concentrations (from 1 ng/mL to 500 ng/mL). 

Moreover, 4-HPR metabolites were identified via HRMS, using an Orbitrap instrument operating in ESI positive ion mode. Subsequently, the transitions identified were used for the quantitation of metabolites.

A description of the similarities and differences between this method and the other available methods [10,12,13,14] is provided in Appendix A. Unlike other methods, this method is able to measure the analytes in the dual plasma and tumor matrix. In addition to this finding, the main advantages of the present method are that a wider range of linearity reduces the need for dilutions and introduced the possibility for the combined assessment of fenretinide and main metabolites allowing for a more comprehensive pharmacokinetic analysis and limiting the required amounts of matrices, at only 30 µL of plasma and 25 mg of tumor. 

The new method was successfully applied to preliminarily assess the pharmacokinetics and bioavailability of BNF, a new oral nanoformulation of 4-HPR, in CD1 mice following acute and chronic oral treatment compared with the free drug. In addition, for the first time, it determined the tumor distribution of 4-HPR and metabolites in a limited number of xenograft models (i.e., melanoma and lung cancer). 

The results of these evaluations indicated that 4-HPR was rapidly available in BNF in vivo and chronic oral treatment with daily administration allowed the achievement of a concentration potentially more effective and superior to those obtained after treatment with free 4-HPR. 

Furthermore, there was no evidence of plasma accumulation of 4-HPR during chronic treatment, i.e., making the treatment well-tolerated. The treatment allowed 4-HPR to penetrate the tumor tissue at levels of the same order of magnitude found in plasma, and most interesting, at concentrations superior the IC50 value of in vitro inhibitory activity. It is also important to emphasize that we observed the presence of comparable tumor levels of the active metabolite 4-oxo-4-HPR, so raising the possibility that the reported in vivo effects of BNF might also depend on the conversion to this metabolite in vivo [18,19,20]. All this leads to a significant exceeding of the aforementioned inhibition activity threshold.

The pharmacokinetic results and preliminary data on intratumoral distribution of 4-HPR reported in this study allowed for the planning of activity studies, which were successfully conducted in several xenograft models [6]. In particular, the plasma and tumor exposure to active concentrations of the drug allowed for the planning of daily treatments with BNF for long periods of time (2 weeks), given also the good tolerability observed in this study.

Considering the positive data originated by the combined exposure of the 4-HPR and metabolite, the newly developed HPLC-MS/MS method appears particularly useful for investigating extensively the in vivo pharmacokinetics of BNF and after validation in human matrices, to be applied in future clinical pharmacokinetic studies.

## Figures and Tables

**Figure 1 pharmaceutics-16-00387-f001:**
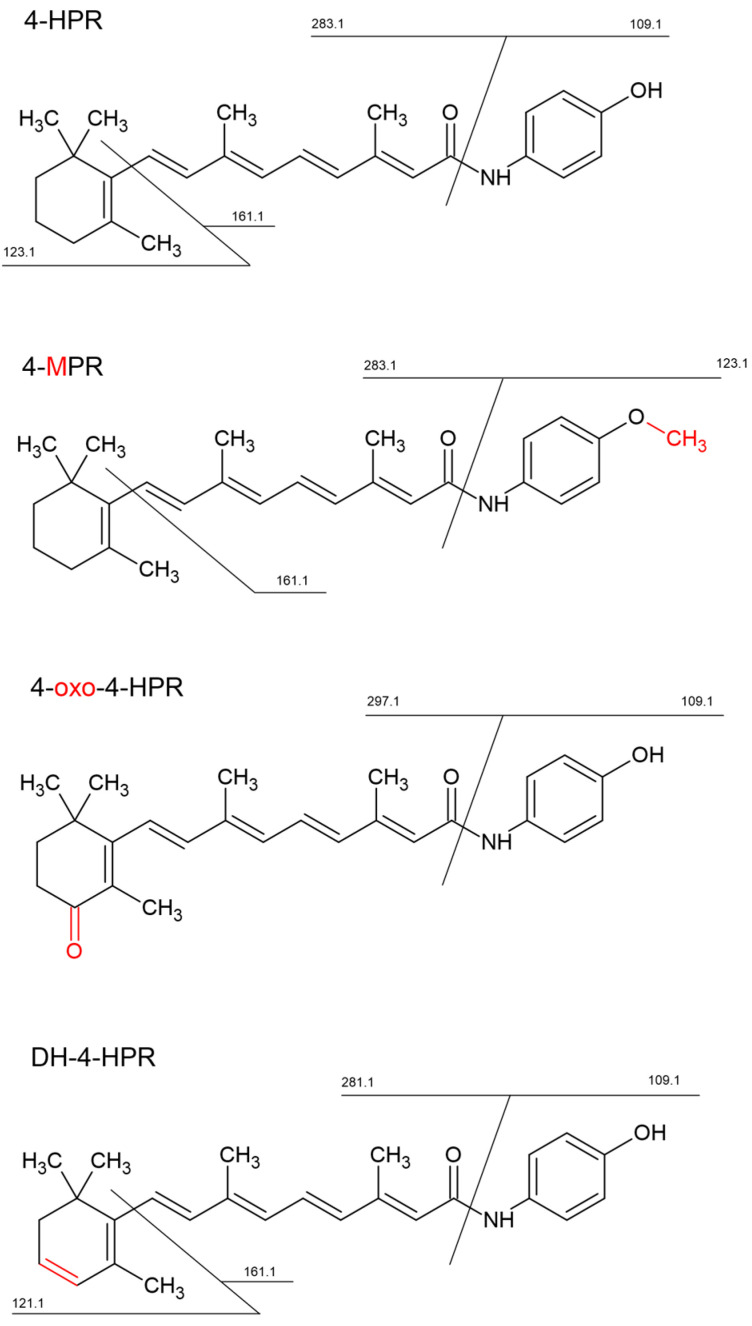
Chemical structure of fenretinide (4-HPR) and main metabolites: 4-MPR (O-methylated), 4-oxo-4HPR (4-oxo-substituted β-ionone ring), DH-4HPR (dehydrogenated 4-HPR). Metabolically modified parts of the structures are highlighted in red.

**Figure 2 pharmaceutics-16-00387-f002:**
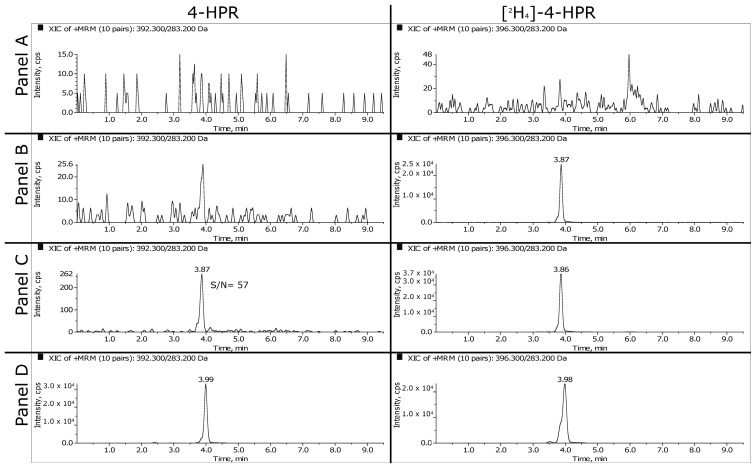
MRM chromatograms of mouse plasma. (**A**) Double blank sample; (**B**) blank sample with IS; (**C**) LLOQ at 1 ng/mL; (**D**) fenretinide and IS of an extracted sample taken 2 h after the last daily dose of BNF (100 mg/kg). The measured concentration corresponds to 1669 ng/mL (sample was analyzed diluted 1:1 with control plasma).

**Figure 3 pharmaceutics-16-00387-f003:**
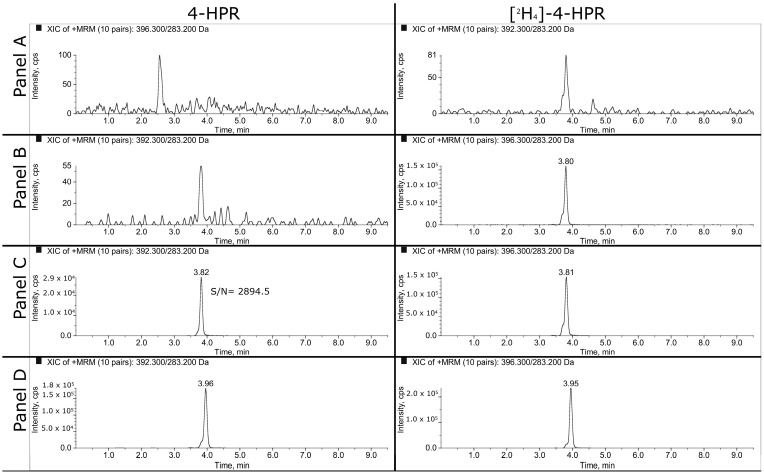
MRM chromatograms of mouse tumor tissues. (**A**) Double blank sample; (**B**) blank sample with IS; (**C**) LLOQ at 50 ng/mL; (**D**) fenretinide and IS of an extracted sample taken 2 h after the last daily dose of BNF (100 mg/kg). The measured concentration in tumor (MEL 3) corresponds to 13,867 ng/g.

**Figure 4 pharmaceutics-16-00387-f004:**
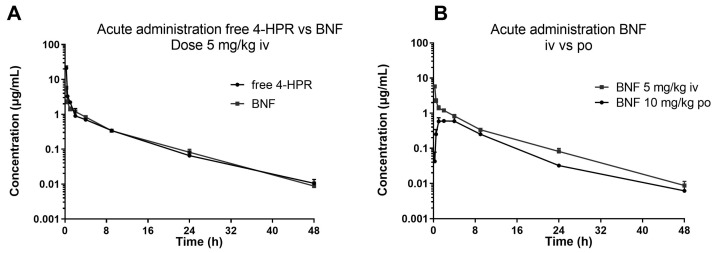
Plasma concentration profiles versus time of 4-HPR after intravenous BNF and free 4-HPR administration at 5 mg/kg 4-HPR equivalent dose (**A**) and oral BNF administration at 10 mg/kg 4-HPR equivalent dose compared with intravenous BNF administration at 5 mg/kg 4-HPR equivalent (**B**).

**Figure 5 pharmaceutics-16-00387-f005:**
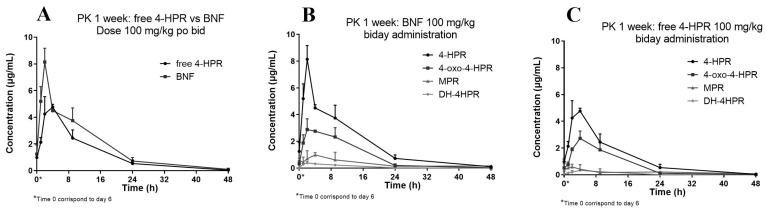
Pharmacokinetic profile of 4-HPR obtained in plasma of mice after 1 week of oral administration of BNF compared to free 4-HPR administration (**A**), pharmacokinetic profile of 4-HPR and metabolites after BNF administration (**B**) pharmacokinetic profile of 4-HPR and metabolites after free 4-HPR administration (**C**).

**Table 1 pharmaceutics-16-00387-t001:** Interday linearity, accuracy and precision of calibration curves of fenretinide prepared in plasma (part (**A**)) or tumor homogenate (**B**).

(A) Plasma 4-HPR Concentration (ng/mL)											
Nominal Concentration (ng/mL)									Calibration Curve		
	1	5	10	25	50	100	250	500	Intercept	Slope	*R* ^2^
Day 1	1.02	5.05	9.92	23.88	52.48	96.14	256.10	503.90	−0.001	0.008	0.9992
Day 2	1.01	4.81	10.25	24.95	50.39	102.60	255.20	477.70	0.000	0.007	0.9996
Day 3	1.01	5.02	9.24	25.24	47.91	100.30	254.20	539.00	0.000	0.008	0.9989
Day 4	1.02	4.55	9.58	25.26	49.73	102.40	260.00	520.30	−0.001	0.008	0.9989
Day 5	1.01	4.78	10.22	23.87	48.59	103.58	253.70	518.24	−0.001	0.007	0.9993
Mean (N = 5)	1.01	4.84	9.84	24.64	49.82	101.00	255.84	511.83	−0.001	0.008	0.99918
SD	0.01	0.20	0.43	0.71	1.77	2.97	2.50	22.80	0.0004	0.0003	0.0003
Precision (%)	0.83	4.13	4.40	2.88	3.56	2.94	0.98	4.45			
Accuracy (%)	101.3	96.8	98.4	98.6	99.6	101.0	102.3	102.4			
**(B) Tumor 4-HPR Concentration (ng/mL)**											
**Nominal Concentration (ng/mL)**							**Calibration Curve**				
	**50**	**100**	**250**	**500**	**1000**	**2000**	**Intercept**	**Slope**	** *R* ^2^ **		
Day 1	50.02	100.57	245.98	495.24	1018.50	2002.10	−0.012	0.002	0.9999		
Day 2	50.33	100.06	237.33	514.37	999.17	2031.10	−0.009	0.003	0.9994		
Day 3	49.66	102.49	237.32	530.11	964.49	2015.70	−0.014	0.003	0.9988		
Day 4	51.49	94.66	245.00	503.32	971.29	2131.30	−0.015	0.003	0.9986		
Mean (N = 4)	50.38	99.45	241.41	510.76	988.36	2045.05	−0.012	0.003	0.99918		
SD	0.79	3.36	4.73	15.10	25.08	58.71	0.0026	0.0007	0.0006		
Precision (%)	1.57	3.37	1.96	2.96	2.54	2.87					
Accuracy (%)	100.8	99.4	96.6	102.2	98.8	102.3					

**Table 2 pharmaceutics-16-00387-t002:** Intraday precision and accuracy for the measure of 4-HPR in plasma.

	Plasma			
	Nominal Concentration (ng/mL)			
	1	8	80	400
Intraday				
	Measured Concentration			
Day 1	1.02	7.29	71.13	357.70
	1.08	6.93	71.45	362.90
	1.08	6.82	71.21	348.70
	0.99	7.83	71.05	360.70
	1.05	7.35	75.61	368.90
Mean (N = 5)	1.05	7.24	72.09	359.78
SD	0.04	0.40	1.97	7.43
Precision (%)	3.64	5.48	2.74	2.07
Accuracy (%)	104.6	89.7	90.1	89.9
	Nominal Concentration (ng/mL)			
		8	80	400
Interday				
		Measured Concentration		
Day 1		7.95	83.70	418.90
		8.54	85.59	429.70
		8.27	83.97	422.80
Day 2		7.88	79.15	385.60
		8.02	89.30	382.00
		8.15	80.78	432.30
Day 3		7.88	79.15	385.60
		8.02	89.30	382.00
		8.15	80.78	432.30
Day 4		8.46	83.99	424.00
		8.78	88.70	421.80
		8.77	87.37	432.30
Day 5		8.10	81.46	388.83
		8.11	80.78	416.33
		8.59	81.50	388.90
Mean (N = 20)		7.99	80.80	397.11
SD		0.55	6.08	28.71
Precision (%)		6.86	7.53	7.23
Accuracy (%)		99.9	101.0	99.3

**Table 3 pharmaceutics-16-00387-t003:** Intraday precision and accuracy for the measure of 4-HPR in tumor.

	Tumor			
	Nominal Concentration (ng/mL)			
	50	150	750	900
Intraday				
	Measured Concentration			
Day 1	50.11	153.64	782.42	946.53
	54.68	154.76	815.42	940.77
	54.27	157.04	776.70	981.73
	55.52	155.51	789.85	976.63
	50.04	153.39	813.89	969.83
Mean (N = 5)	52.92	154.87	795.66	963.10
SD	2.64	1.49	17.97	18.36
Precision (%)	4.99	0.96	2.26	1.91
Accuracy (%)	105.8	103.3	106.1	107.0
	Nominal Concentration (ng/mL)			
Interday		150	750	900
		Measured Concentration		
Day 1		150.34	765.33	939.23
		150.75	779.29	946.53
		155.54	743.97	933.29
		154.22	756.63	935.50
		149.82	753.52	921.24
		150.91	755.05	932.49
Day 2		149.62	796.99	986.29
		154.88	788.64	974.25
		150.35	807.83	1000.80
Day 3		154.34	766.09	905.20
		157.20	754.69	911.21
		150.03	775.60	986.18
Day 4		152.64	811.27	936.07
		152.56	748.20	1002.00
		160.44	776.32	911.66
Mean (N = 20)		153.40	777.89	951.87
SD		2.93	22.72	30.40
Precision (%)		1.91	2.92	3.19
Accuracy (%)		102.3	103.7	105.8

**Table 4 pharmaceutics-16-00387-t004:** Pharmacokinetic parameters of 4-HPR and metabolites after repeated administration of oral BNF (**A**) and free 4-HPR (**B**).

(A) BNF						
Parameter	Cmax ± SD (µg/mL)		Tmax (h)	AUC_0–last_ (h*µg/mL)	R% *	HL (h)
4-HPR	8.14	±1.029	2	86.28	-	7.5
oxo-4-HPR	2.88	±0.809	2	43.40	50%	6.1
MPR	0.98	±0.179	4	14.82	17%	16.6
DH-4-HPR	0.38	±0.125	2	4.73	5%	6.2
**(B) Free 4-HPR**						
**Parameter**	**Cmax ± SD** **(µg/mL)**		**Tmax** **(h)**	**AUC_0–last_** **(h*µg/mL)**	**R% ***	**HL** **(h)**
4-HPR	4.76	±0.205	4	60.94	-	6.6
oxo-4-HPR	2.72	±0548	4	34.30	56%	5.3
MPR	0.59	±0.193	2	9.60	16%	21.7
DH-4-HPR	0.33	±0.068	4	4.09	7%	5.3

* R = AUC_0–last_ metabolite/AUC_0–last_ 4-HPR.

## Data Availability

The data supporting the reported results are available from the corresponding author upon request.

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
