# Peer review of "Validated LC-MS/MS Assay for the Quantitative Determination of Fenretinide in Plasma and Tumor and Its Application in a Pharmacokinetic Study in Mice of a Novel Oral Nanoformulation of Fenretinide"

_pharmaceutics, 2024, doi:10.3390/pharmaceutics16030387_

Round 1

Reviewer 1 Report

Comments and Suggestions for Authors

The manuscript presents a thorough work on a new orally absorbed nanoformulation of fenretinide. I consider the analytical description and planning of the study to be excellent, the formulated goals are well defined, the presentation of the results and the conclusions drawn comply with the rules of the discipline. That being said, I would suggest a few fixes, in order of appearance and in general, as follows:

- the term tumor / tumour appears quite heterogeneously in the text, it would be worthwhile to standardize this.

- line 41: when entering the name of the active ingredient, N (nitrogen) should be entered in italics.

- in Fig1, it would be worthwhile to mark the metabolically sensitive point in the structure of the individual metabolites, or to specify the type of metabolism after the abbreviation (e.g. 4-MPR (O-methylated)

- line54: I could not find an explanation for the abbreviation RARb.

- line82: A space before the word In is missing

- tumor tissue samples are mentioned in several places in the manuscript, but in my opinion, this is not accurately described even in the methodological description. Exactly what kind of tissue sample are we talking about, or what cell line(s), where were the samples taken from, how is the tumor control homogenate defined?

- two independent MS techniques are used (QQQ / Orbitarp) with different ionization techniques (ESI / APCI). In my opinion, this is unusual. It would be worthwhile to explain the reason for choosing the two ionization techniques, as well as the differences in sensitivity related to the active ingredient in relation to ESI / APCI.

- In line 277, I think the abbreviation SRM is incorrectly stated, the presented results (Fig2) also refer to MRM, this should be corrected.

Comments on the Quality of English Language

I have no further comments for quality of English.

Author Response

Dear Referee

Below we provide you with the answers to your welcome comments and suggestions.

Additionally, we have uploaded the word version of the article with the revisions marked and with some parts of the text reworded and the number of the references corrected as requested by the Editor.

Points 1 and 2.

According to the Referee's requests we use exclusively the term "tumor" and N (nitrogen) is now written in italics.

Point 3.

Figure 1 has been modified as suggested by the Referee, marking in red the point of the structures where the metabolic modification occurred and specifying the type of metabolism in the legend.

Points 4 and 5.

The Referee's requests were implemented through the text

Point 6.

The tumors used in the “application of the method” section and as control, were generated by the subcutaneous inoculation of cancer stem cells of different types. The explanation is now included at start of paragraph.2.5.

Point 7.

The use of two types of MS instruments depended on the fact that, as indicated in paragraph 2.7 and 3.2, initially by mean of a high-resolution mass spectrometer (LTQ-Orbitrap XL) we qualitatively identified the metabolites present in "test samples of plasma and tumor" and defined the M/Z transitions to be monitored and quantified with the API 4000 instrument (QQQ). The use of the APCI source in the QQQ is due to the fact that 4-HPR ionizes better with this source than with ESI.

Point 8

According to the referee's comment, the abbreviation SRM has been corrected to MRM (Multiple Reaction Monitoring).

Reviewer 2 Report

Comments and Suggestions for Authors

The Authors developed the LC-MS/MS method for determination of fenretinide. It was applied in the pharmacokinetic study. Having read the manuscript the following questions appeared:

1. How many animals were in the investigated groups of mice, and what was the volume of the sample taken? I presume that it was 30 microliters (line 85). This information should be in 2.8.5. paragraph.

2. What was the mean body weight of the animals in each of group?

3. In line 132 I would add the volume of the added IS rather than the quantity (3 ng).

4. Were the equations for the calibration curve tested to exclude the quadratic trend? The calibration curve range is broad and there is a possibility that the quadratic function could also fit.

5. Was the method optimized? Was the robustness checked?

6. The Authors claimed that validation obey the EMA recommendations. In the latest recommendations the precision and accuracy should be checked for 4 concentrations i.e.: LLOQ, 3*LLOQ (low QC), 30-50% calibrationn range (medium QC), at least 75% of the ULOQ (high QC). In my opinion it should be checked.

7. In table 4 I would use t0.5 instead of HL (I suppose it was half-life).

8. According to what model were the pharmacokinetic parameters were calculated? One- , two- or non-compartmental analysis? What software was used for the determination of pharmacokinetic parameters?

9. Was the method tested towards the potentially co-administered drugs?

10. What is the therapeutic index for fenretinide? Is it known?

Author Response

Dear Referee

Below we provide you with answers to your welcome comments and suggestions.

Additionally, we have uploaded the final word versions of the article with the revisions marked and some parts of the text reworded and the number of references corrected as requested by the Editor.

Point 1-2

As Requested by the referee we specified in 2.8.5 the numbers of mice utilized 48 (N=24 for each group) in the pk study and per time point (N=3). We also specified the mean weight of mice (25 g) and the volume of plasma collected (250-300 µl). 30 µL was the volume of sample assayed.

Point 3

The information on the volume of IS (10 µL) added to sample was included in paragraph 2.4.

Point 4-5, 9

The quadratic equation was not tested, to tell the truth the linear one performed very well (R>0.999) in the applied dynamic range of concentrations, so we did not try testing different equations.

The method had been optimized based on our previous publication (Sala et al, Journal of Chromatography B, 2009: 877, 3118-3126) in the field of retinoic acid derivatives. The robustness, however, was not tested in depth, but by carrying out a change of matrix as reported plasma/tumor (2 types).

Influences on the method of concomitant medications have not been verified.

Point 6

We agree with Referee, the precision and accuracy at 4 QCs level of concentration, including LLOQ, were checked in the within run analysis. We have now included the data in revised version of tables 3 and 4.

Point 7,8

Table 4 has been modified as suggested by the Referee and the information about PK data processing way (using a non-compartmental model) has been included in paragraph 2.8.5.

Point 10

Although the therapeutic index of fenretinide is not precisely known, the safety information about its intense chronic use (ref 12) without the evidence of particular signs of intolerability suggests a good therapeutic index.

Round 2

Reviewer 1 Report

Comments and Suggestions for Authors

All the problematic points I raised were properly addressed in the manuscript. I recommend publication in its present form.

Reviewer 2 Report

Comments and Suggestions for Authors

The introduced modifications improved the quality of the manuscript.